METHODS AND RESOURCES

# A flexible framework for multi-particle refinement in cryo-electron tomography

**Alister Burt**[1]*, **Lorenzo Gaifas**[1], **Tom Dendooven**[2], **Irina Gutsche**[1]*

**1** Institut de Biologie Structurale, Université Grenoble Alpes, CEA, CNRS, IBS, Grenoble, France, **2** MRC Laboratory of Molecular Biology, Cambridge, United Kingdom

* alisterburt@gmail.com (AB); irina.gutsche@ibs.fr (IG)

## Abstract

Cryo-electron tomography (cryo-ET) and subtomogram averaging (STA) are increasingly used for macromolecular structure determination in situ. Here, we introduce a set of computational tools and resources designed to enable flexible approaches to STA through increased automation and simplified metadata handling. We create a bidirectional interface between the *Dynamo* software package and the *Warp-Relion-M* pipeline, providing a framework for ab initio and geometrical approaches to multiparticle refinement in *M*. We illustrate the power of working within this framework by applying it to *EMPIAR-10164*, a publicly available dataset containing immature HIV-1 virus-like particles (VLPs), and a challenging in situ dataset containing chemosensory arrays in bacterial minicells. Additionally, we provide a comprehensive, step-by-step guide to obtaining a 3.4-Å reconstruction from *EMPIAR-10164*. The guide is hosted on https://teamtomo.org/, a collaborative online platform we establish for sharing knowledge about cryo-ET.

## Introduction

Cryo-electron tomography (cryo-ET) is an imaging technique rapidly gaining popularity for the direct visualisation of unique biological objects in 3D. Repeating structural motifs present in cryo-ET data can be reconstructed at higher resolution by subtomogram averaging (STA), allowing for the possibility of studying macromolecular structure in situ [1]. STA has developed alongside single-particle analysis (SPA) in cryo-electron microscopy (cryo-EM), a technique that has benefitted significantly in the last 10 years from advances in both electron detection hardware and image processing software [2,3].

State-of-the-art STA workflows often co-opt tools and adapt ideas from SPA for tomography and STA [4,5]. An unfortunate consequence of this side-by-side development is a somewhat fragmented software ecosystem with no standardisation of file formats or metadata conventions [6]. Despite the advent of complete or near-complete integrated solutions for cryo-ET and STA [7–9], optimal methodology for a given dataset often requires the creative combination of different approaches that may not all be present within a single integrated pipeline. For those new to cryo-ET, the burden of interfacing many different software packages in a complex workflow often represents a barrier to the testing of alternative approaches.

Image processing for STA from cryo-ET data is a complex, multistep process that can broadly be divided into 3 blocks (Fig 1). The first, "preprocessing," generates 3D

**Data Availability Statement:** Data on immature HIV-1 dMACANC virus-like particles used for benchmarking of the pipeline was taken from EMPIAR-10164. Data on E. coli minicells will be available upon publication of the final

reconstruction of the chemosensory array. Information underlying data displayed at Figs 5 and 6 is available for download at doi:10.5281/zenodo.4783129 and doi:10.5281/zenodo.4783151 respectively. Source code for all tools and resources described here is available at: • autoalign_dynamo - https://github.com/alisterburt/autoalign_dynamo • mdocspoofer - https://github.com/alisterburt/mdocspoofer • starfile - https://github.com/alisterburt/starfile • dynamotable - https://github.com/alisterburt/dynamotable • eulerangles - https://github.com/alisterburt/eulerangles All Python packages are made available on the Python package index (PyPI). Benchmarking code for eulerangles can be found at https://gist.github.com/alisterburt/4a32e9c122498ac0ab482ee5ba44ba10.

**Funding:** This work has received funding from a European Union's Horizon 2020 research and innovation programme under grant agreement No 647784 to IG. AB is supported by a University Grenoble Alpes PhD fellowship and by a Fondation pour la Recherche Medicale (FRM) fellowship FDT202001011069. LG is supported by a PhD fellowship from Grenoble Alliance for Integrated Structural and Cell Biology (GRAL, ANR-10-LABX-49-01) funded within the CBH Graduate School of the University Grenoble Alpes (ANR-17-EURE-0003). The funders had no role in study design, data collection and analysis, decision to publish, or preparation of the manuscript.

**Competing interests:** The authors have declared that no competing interests exist.

**Abbreviations:** API, application programming interface; cryo-EM, cryo-electron microscopy; cryo-ET, cryo-electron tomography; CSU, core signalling unit; CTF, contrast transfer function; EMDB, Electron Microscopy Data Bank; FIB, focused ion beam; FSC, Fourier shell correlation; GUI, graphical user interface; PCA, principal component analysis; RMSD, root mean square deviation; SPA, single-particle analysis; STA, subtomogram averaging; VLP, virus-like particle.

reconstructions (tomograms) from experimental 2D micrographs. The second, "particle picking," creates putative positions and orientations for objects of interest within each volume as well as initial reference(s) for subsequent refinement. Finally, the "refinement" block is concerned with the optimisation of reconstruction(s) from imaging data associated with a set of particle positions.

The preprocessing block encompasses all steps in the generation of tomograms from experimental data. Experimental cryo-ET data are usually acquired as a set of multiframe 2D micrographs, one per tilt angle in a tilt series. Typically, this block includes per tilt interframe motion estimation and correction, contrast transfer function (CTF) estimation, tilt series alignment, and 3D reconstruction [10]. Tilt series alignment is the least automated of these steps, with significant time often spent optimising alignments in an attempt to produce more accurate reconstructions.

Particle picking is somewhat entangled with the process of initial model generation, with some particle picking methods requiring an initial model and others being reference free. Exhaustive, reference-based template matching approaches are widely accepted as an imperfect solution for particle picking due to the computational overhead and the need for subsequent dataset cleaning [11,12]. Care must also be taken to avoid model bias. Particle picking methods based on deep learning have been proposed, which may help to address some of these limitations [13,14]. Another class of particle picking methods, termed "geometrical particle picking," derives putative particle positions and orientations from a 3D model of a supporting geometry, usually generated from minimal manual annotations [15]. These methods can be used to impose prior knowledge about how particle poses relate to the supporting geometry during refinement. Employing such a priori knowledge is advantageous, reducing both the computational burden of global searches and the likelihood of ending up in incorrect local minima during refinement. These advantages come at the expense of extra time spent on manual annotation and metadata management.

Refinement in STA has typically meant the optimisation of particle poses, considering particles as rigid bodies within fixed 13D reconstructions [16]. The potential for studying macromolecular structure at intermediate resolution (3 to 5 Å) by STA became reality with the advent of an efficient method for 3D CTF correction [17]. Inspired by progress in SPA, the refinement block may now include reference-based procedures for the a posteriori optimisation of tilt series alignment parameters, frame series alignments, and electron-optical parameters [7–9,18]. Despite these algorithmic advances, there remain relatively few examples of reconstructions from tilt series data in this resolution regime in the Electron Microscopy Data Bank (EMDB), indicating that this work remains nontrivial (S1 Fig). We posit that a possible reason for this discrepancy is the complexity of integrating the various tools available for each step of the workflow in the fragmented cryo-ET software ecosystem.

*Warp-RELION-M* is a user-friendly solution for cryo-ET preprocessing, STA, and multi-particle refinement, yielding the highest-resolution reconstructions from cryo-ET data seen thus far [9]. The current version of *Warp (*v1.0.9) is designed to process data collected using *SerialEM* and does not integrate automated tilt series alignment procedures. The pipeline assumes that an initial model is available for template matching and does not interface directly with tools for geometrical particle picking. *Dynamo* is a flexible, extensible software environment for STA, providing a variety of powerful tools for the management, annotation, and analysis of cryo-ET data [15,19].

The combination of the tools in these software packages would yield a computational framework capable of more automated, ab initio, and geometrical approaches to multi-particle refinement in *M*. In this manuscript, we describe our efforts to facilitate working within this framework. In the Materials and methods section, a set of tools is presented, which automate fiducial-based tilt series alignment, simplify metadata handling, and create a bidirectional

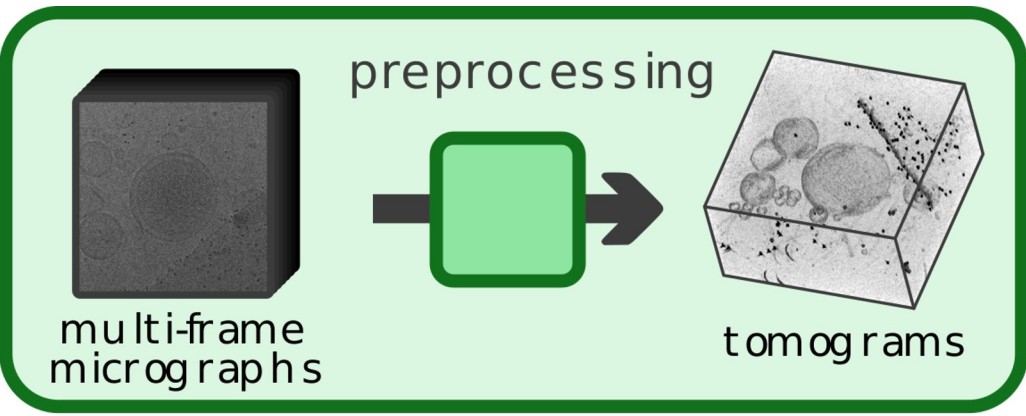

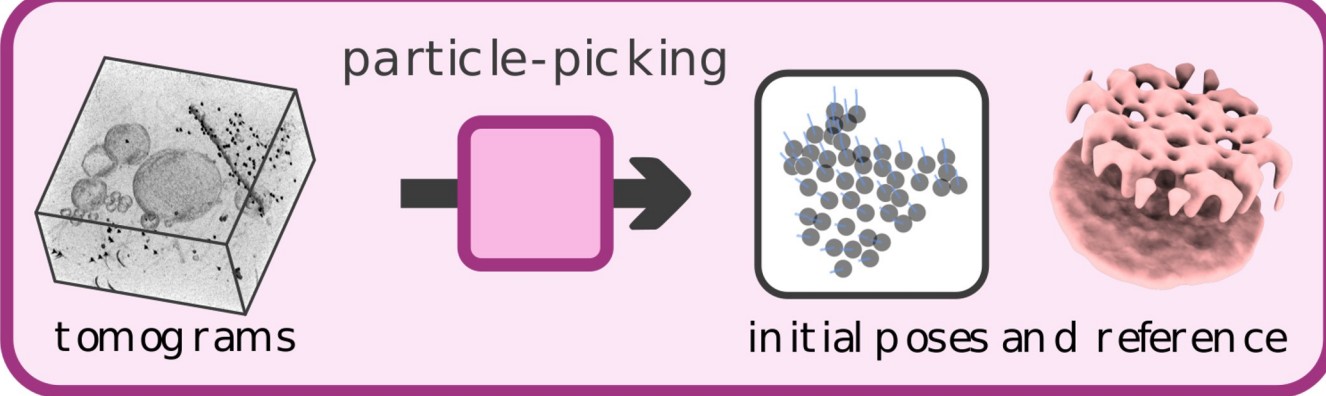

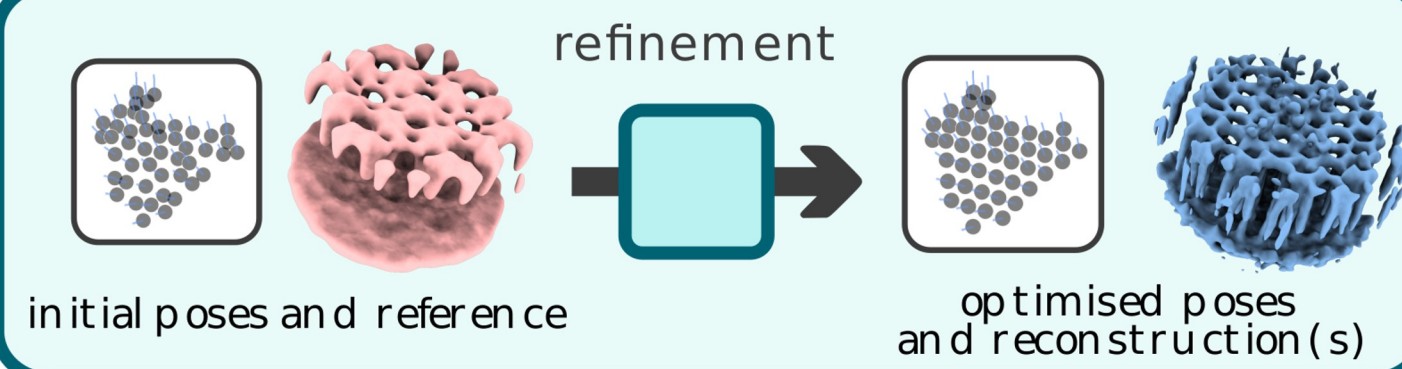

**Fig 1. Cryo-ET data processing can be roughly divided into 3 blocks.** Preprocessing (top) transforms raw experimental data, usually in the form of multiframe micrographs, into 3D images called tomograms. Particle picking (middle) derives particle position and orientation information from tomograms as well as an initial reference for subsequent refinement. Refinement (bottom) optimises (multiple) 3D reconstructions from input data. cryo-ET, cryo-electron tomography.

interface between *Dynamo* and *Warp-RELION-M*. In the Results and discussion section, we demonstrate the benefits of our metadata handling tools and illustrate the power of working within this computational framework on 2 datasets. Finally, we provide a comprehensive step-by-step guide to obtaining a 3.4-Å reconstruction of the HIV-1 CA-SP1 hexamer ab initio from 5 tilt series of *EMPIAR-10164*. This guide is presented as a part of a collaborative, online platform (https://teamtomo.org/) that we establish as a platform for sharing knowledge about cryo-ET data processing.

## Materials and methods

### Integrating on-the-fly tilt series alignment into the Warp preprocessing pipeline

Tomogram reconstruction within the *Warp* preprocessing pipeline currently requires the use of an external software package, *IMOD* [20], for tilt series alignment. In the absence of a fully integrated solution in the currently available version of *Warp* (1.0.9), we developed *autoalign_dynamo*, a package for automating fiducial-based tilt series alignment using *Dynamo* and *IMOD*. *dautoalign4warp* is a program for on-the-fly alignment of batches of tilt series from *Warp*. The program proceeds by dynamic generation of a *Dynamo* tilt series alignment workflow with appropriate parameters derived from minimal user input. The workflow is executed, generating a set of refined fiducial positions. Refined fiducial positions are converted using *dms2mod* (provided by *autoalign_dynamo*) and used to generate tilt series alignment parameters using the *IMOD* program *tiltalign*. Key *tiltalign* parameters fixed within this procedure are the following: solve for 1 tilt axis for the entire tilt series; fix tilt angles at their nominal values; and use robust fitting for parameter estimation. All *tiltalign* parameters can be seen and modified in the *tiltalign_wrapper* script. All data are subsequently organised such that alignments can be imported directly into Warp for tomogram reconstruction, requiring no further user input. The base function *autoalign*, which wraps *Dynamo* tilt series alignment functionality, is provided for those who wish to apply this procedure outside of the *Warp* pipeline.

Dynamo tilt series alignment workflows were released recently, and, as such, are not yet described in the literature, although details and the source code are available in the public domain [21]. To facilitate the readers understanding, we provide a brief overview of the alignment algorithm, noting that it was not implemented by us. A binary, synthetic template of a fiducial marker is generated based on user input and used to detect candidate fiducial positions within a tilt series by cross-correlation. A total of 300 subimages are extracted from the tilt series at the peaks of the resulting cross-correlation matrix and averaged to produce a template for detecting fiducial markers in the data. The template is used to detect initial fiducial positions in the tilt series by cross-correlation. Cross-correlation peaks are analysed, and those observations not meeting a minimum degree of rotational symmetry are discarded. Observations are indexed to link observations of the same fiducial marker in multiple micrographs by pairwise cross-correlation of the observations between neighbouring micrographs. The longest "trails" of linked observations are used to generate an initial 3D model of fiducial positions. The 3D model is iteratively reindexed by reprojection of fiducial positions against the tilt series, adding observations that fall within a distance threshold to the set of observations to be included for further refinement. Tilt images lacking at this point are reintegrated by a procedure comparing the reprojection of the current model with the cloud of initial observations found on that micrograph. The number of observations is maximised by the reintegration of missing fiducial markers from other images using the same procedure. The projection model is iteratively refined before the positions of fiducial markers are independently iteratively refined against an average of all observations of that marker. The final set of refined markers is pruned according to the root mean square deviation (RMSD) between measured fiducial position and the reprojected position of the 3D model.

### A set of self-contained Python packages for metadata handling

In an effort to interface the primary metadata systems of *RELION* [22] and *Dynamo* [19] with the scientific Python [23] ecosystem, we provide 2 Python packages: *starfile* and *dynamotable*. These packages provide input/output functionality via a simple application programming interface (API), exposing data as *pandas* DataFrame objects.

Euler angles are used by many cryo-EM software packages to describe the orientation of a rigid body with respect to a fixed coordinate system. *eulerangles* is a Python package that provides a simple API for the batch conversion of Euler angles into rotation matrices, rotation matrices into Euler angles, and interconversion of Euler angles defined according to different conventions. Conversions can take place between all possible formulations of Euler angles in a right-handed coordinate system. A simple mechanism is provided for the definition and reuse of conventions from specific software packages. Documentation for *eulerangles* can be found at https://eulerangles.readthedocs.io/.

## Enabling Warp preprocessing for data collected in Tomography 5

In the *Warp* cryo-ET preprocessing pipeline, only metadata from the *SerialEM* data collection program in the form of mdoc files are currently supported. Thermo Scientific *Tomography 5* (Tomo 5) is the official solution provided with Thermo Scientific microscopes for electron tomography experiments [24]. We provide a small command line tool *mdocspoofer* for the generation of mdoc files for a directory containing multiframe micrograph files of the form *<basename>_<count>[<tilt_angle>]_fractions.mrc*, as generated by *Tomography 5*. This tool enables use of the *Warp* preprocessing pipeline for users of *Tomography 5*.

## Interfacing Dynamo and the Warp-RELION-M pipeline

The recommended procedure for particle picking in the *Warp-RELION-M* pipeline is exhaustive, reference-based template matching. *Dynamo* offers an interactive environment for picking particles based on supporting geometries in cryo-electron tomograms [15]. We provide *warp2catalogue*, a program that sets up a database called a *Dynamo* catalogue for tomograms reconstructed in *Warp*. The catalogue is set up such that all visualisation operations make use of a deconvolved reconstruction, filtered for improved visualisation, while subsequent particle extraction uses the corresponding unfiltered volume, simplifying the experience for the user.

For rigid body particle pose optimisation, STA in *RELION* is integrated into the existing pipeline. Alternative approaches to particle picking, pose optimisation, and classification may provide advantages over this workflow. We provide *dynamo2m*, a set of tools that create a bidirectional interface between *Dynamo* and the *Warp-RELION-M* pipeline. *dynamo2warp* allows for particle position and orientation data in Dynamo to be used for particle reconstruction in *Warp*, rigid body refinement in *RELION*, and multi-particle refinement in *M*. *warp2dynamo* provides a route to using particles reconstructed in *Warp* within *Dynamo*. An additional utility, *relion_star_downgrade*, is provided to enable reconstructing particles refined using *RELION* version 3.1 or higher in *Warp* 1.0.9.

# Results and discussion

## Tools for simplified metadata handling

The ability to test alternative approaches and rapidly iterate is key to optimising complex data analysis workflows like cryo-ET and STA. Exploratory data analysis such as checking the internal consistency of particle positions within a supramolecular assembly after a refinement is often key to understanding the limitations of a given approach. Implementing custom input/output functionality when attempting to work with a variety of nonstandard metadata presents a barrier to entry for those wishing to interactively explore their data or implement custom analysis routines. The Python packages *starfile* and *dynamotable* provide simple interfaces between the metadata systems of *RELION*, *Dynamo*, and a wealth of data analysis infrastructure, visualisation tools, and educational resources in the scientific Python ecosystem [25,26].

Converting between different Euler angle conventions is often a pain point for those wishing to interface different cryo-EM software packages due to the abundance of ambiguities in their interpretation [27]. *eulerangles* simplifies generating rotation matrices from Euler angles for custom analyses and facilitates building interfaces between pieces of software that interpret Euler angles according to differing conventions. Our vectorised implementation is approximately 10× faster than the popular scipy utility *scipy.spatial.transform.Rotation*, taking 381 ms versus 4.38 s for the conversion of a million Euler triplets, the *eulerangles* package thus being more suitable for working with large sets of Euler angles in Python.

As an example of the benefits derived from working within this ecosystem, we combine *starfile* and *eulerangles* with the packages *mrcfile* [28] (for reading and writing MRC format image files) and *napari* [29] (a fast, powerful multidimensional data visualisation library) to visualise tomograms, particle positions, and their oriented Z-vectors in 3D (Fig 2). Using these packages, creating such a visualisation is achieved in less than 25 lines of (human readable) code (S2 Fig).

## A flexible framework for ab initio and geometrical approaches to subtomogram averaging

Each STA dataset is different, and analysis presents its own unique challenges. The tools provided in *autoalign_dynamo* and *dynamo2m* were designed to simplify cryo-ET data processing where possible and empower researchers with access to a vast array of alternative tooling where it may provide benefit. Combined, these tools enable working within a powerful, flexible framework for more automated, ab initio, and geometrical approaches to multi-particle refinement, linking *Warp-RELION-M* with *Dynamo* (Fig 3). In this section, we discuss and illustrate the benefits provided by our tools.

## Automation of tilt series alignment in *Warp-RELION-M* with *autoalign_dynamo*

Optimisation of data collection strategies has significantly increased the throughput of cryo-ET data collection in recent years, with datasets now routinely exceeding 100 tilt series. As dataset sizes increase, so does the need for accurate, automated solutions to each and every step of complex data analysis workflows. Accurate tilt series alignment is often achieved using semiautomated procedures that quickly become tedious when faced with a large dataset. Providing the automated, robust fiducial-based tilt series alignment procedures from *Dynamo* as an on-the-fly solution tightly integrated into the *Warp* preprocessing pipeline greatly simplifies the generation of accurate 3D reconstructions for downstream data analysis. Algorithmically, *Dynamo* tilt series alignment differs from *batchruntomo* in *IMOD* in a few significant ways [30]. In *IMOD*, the fiducial center position is determined on a Sobel-filtered image. *Dynamo* iteratively refines the center position of fiducials in each image by alignment to a per-fiducial reference image generated from images of that fiducial in the whole tilt series. Additionally, the *Dynamo* procedure explicitly attempts to maximise the number of observations in a tilt series by reintegrating missing fiducials, leveraging an existing projection model for detection of missing fiducials and subsequent improvement of the model. In contrast, *IMOD* does not attempt to reintegrate markers not included in the initial seed model unless explicitly added by the user. A quantitative comparison between different automated tilt series alignment algorithms could form the basis for interesting future work.

We provide no integrated solution for aligning tilt series lacking exogenous fiducial markers, such as those from samples prepared by focused ion beam (FIB) milling, although procedures exist in other software packages [8]. As new tilt series alignment algorithms are developed, their integration into on-the-fly data processing workflows will be important for optimal use of both microscope and researcher time.

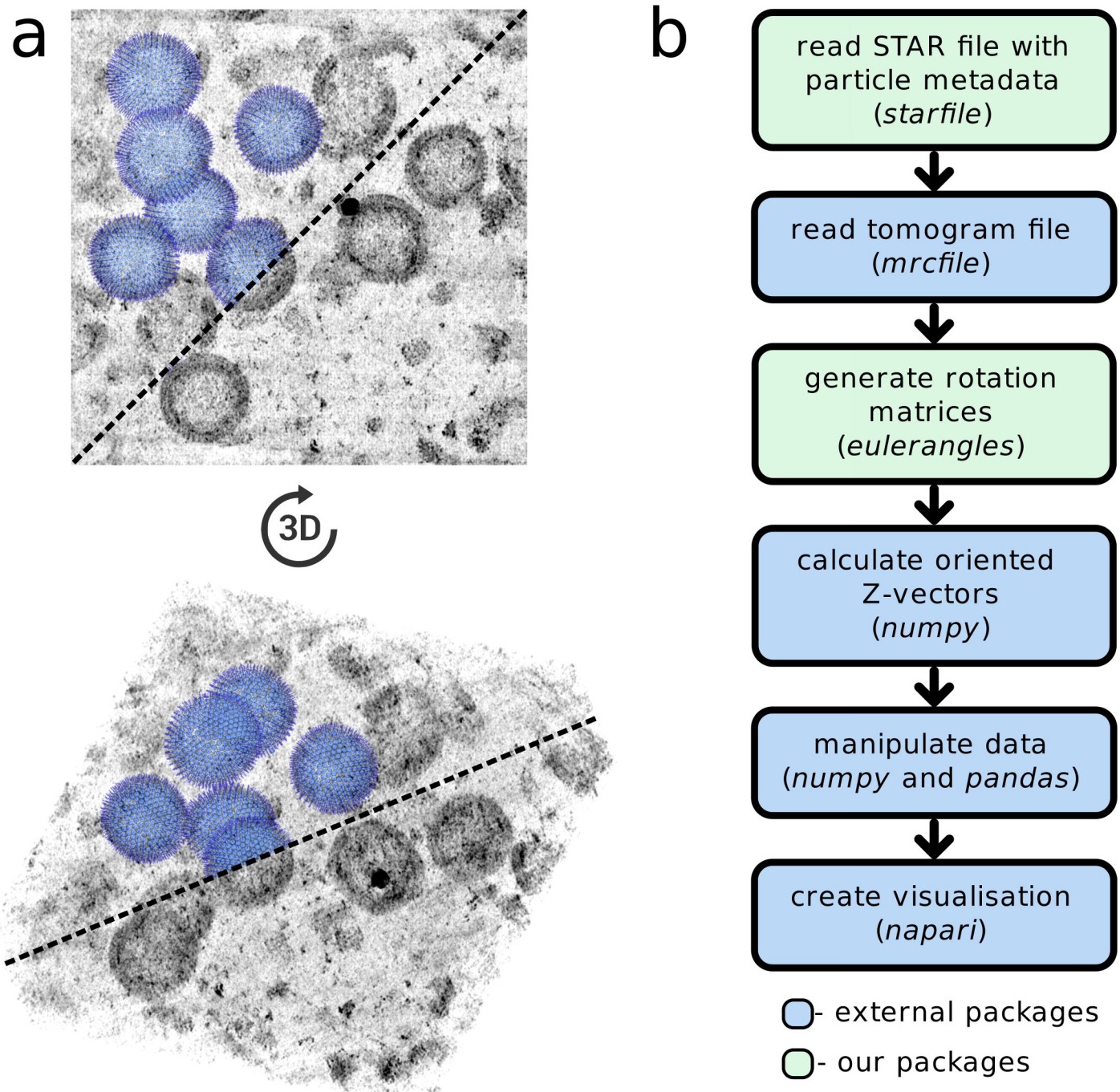

**Fig 2. Modular tools designed to integrate with the scientific Python ecosystem simplify the creation of custom visualisations in *napari*.** (a) Particle positions and orientations, easily accessed and manipulated using *starfile* and *eulerangles*, are rendered in 3D with the imaging data from which they were derived. (b) A flowchart of the process for creating the visualisation in (a) demonstrates the benefits provided by an interface with the scientific Python ecosystem.

### Enabling geometrical approaches to subtomogram averaging in *Warp-RELION-M*

*warp2catalogue* and *dynamo2warp* enable the use of geometrical approaches to STA to users of the *Warp-RELION-M* pipeline (Fig 4). The catalogue system in *Dynamo* and the interactive

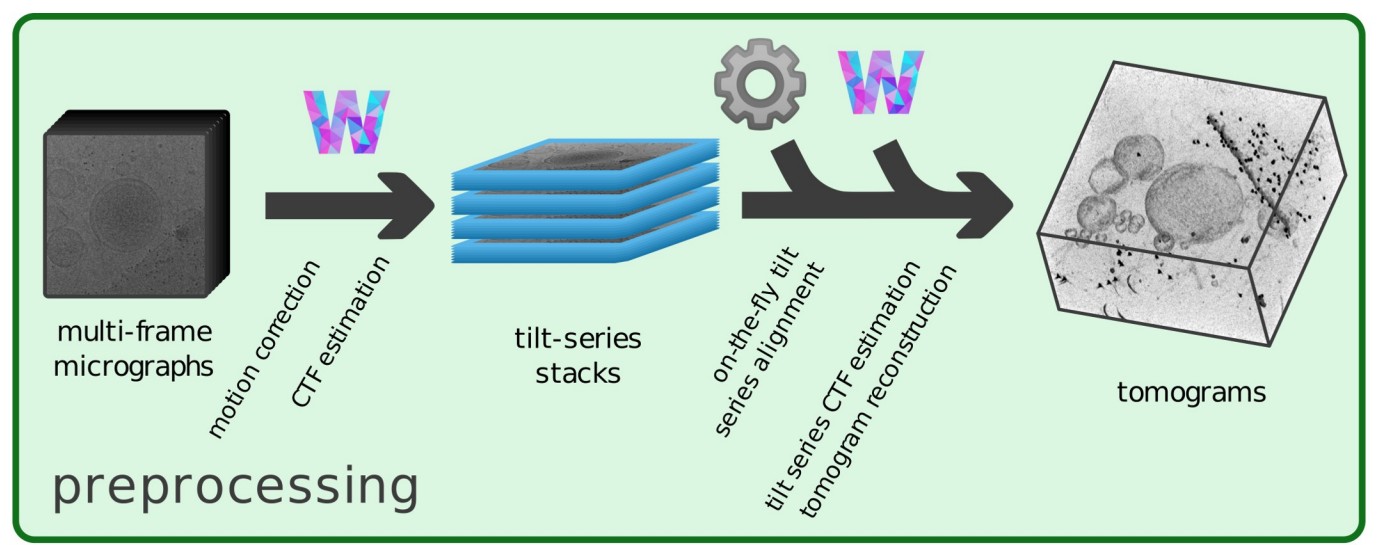

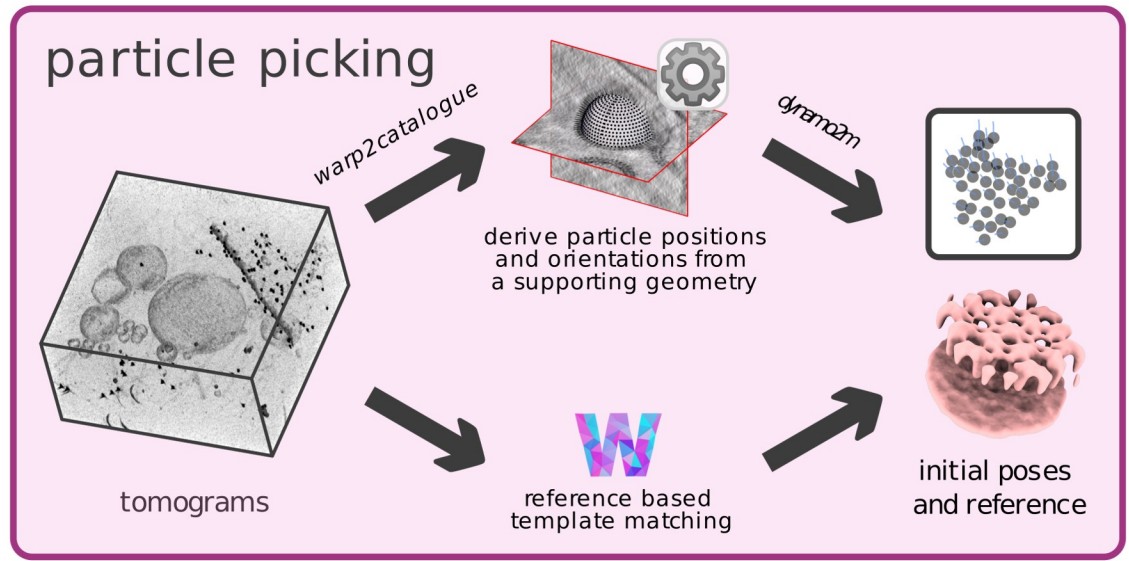

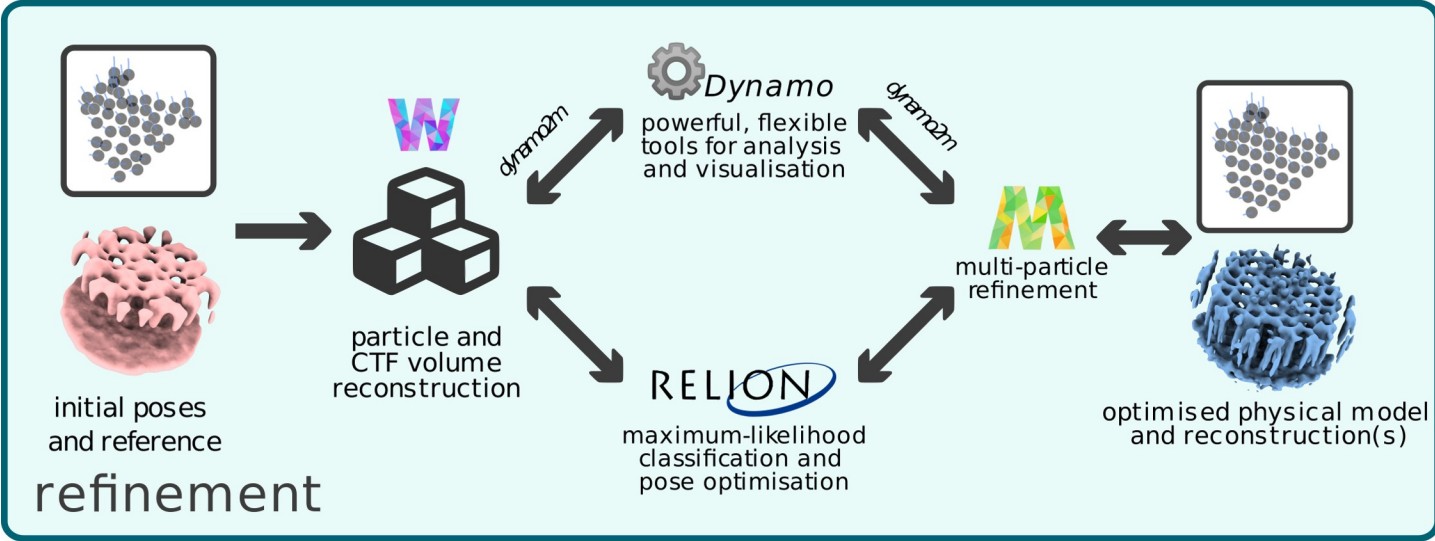

**Fig 3. *autoalign_dynamo* and *dynamo2m* enable a powerful, flexible framework for ab initio and geometrical approaches to multi-particle refinement of cryo-ET data.** The cryo-ET preprocessing pipeline in *Warp* (steps marked by a *W*, top) is extended to include on-the-fly tilt series alignment in *Dynamo* (marked by a grey cog). Users benefit from increased automation. *warp2catalogue* augments the particle picking block in *Warp-RELION-M* to facilitate the use of geometrical particle picking tools in *Dynamo*, enabling ab initio reconstruction leveraging prior knowledge of particle orientation during alignment and averaging. In the refinement block, *dynamo2m* enables users to move freely between *Dynamo* and *Warp-RELION*-M, choosing the tool most well suited to the problems posed by their data. Combined, these tools enable ab initio and geometrical approaches to multi-particle refinement of cryo-ET data. cryo-ET, cryo-electron tomography; CTF, contrast transfer function.

3D modelling tools in the *dtmslice* viewer make it easy to generate models of supporting geometries in sets of tomograms and use them to derive particle positions and orientations for subsequent STA experiments [15]. Examples of easy-to-annotate supporting geometries from which putative particle poses can be easily derived are vesicles (Fig 4A), arbitrarily shaped membrane surfaces (Fig 4B), filaments, and crystals. *warp2catalogue* simplifies the setup of a Dynamo catalogue for *Warp-RELION-M* users: The user annotates volumes filtered to aid visualisation, whereas particle extraction operations performed from the catalogue use the corresponding unfiltered data. This automation simplifies the use of an optimal workflow without unnecessary cognitive burden for the user.

*Dynamo* also provides *dpktbl.subbox.tableOnTable* for "subboxing," deriving particle positions and orientations, which are geometrically related to an existing set of positions and orientations (Fig 4C). This procedure is often useful in cryo-ET for focussing analysis on subunits of a large complex after an initial consensus refinement [31]. *dynamo2warp* provides a simple mechanism for the integration of these powerful tools into the *Warp-RELION-M* pipeline.

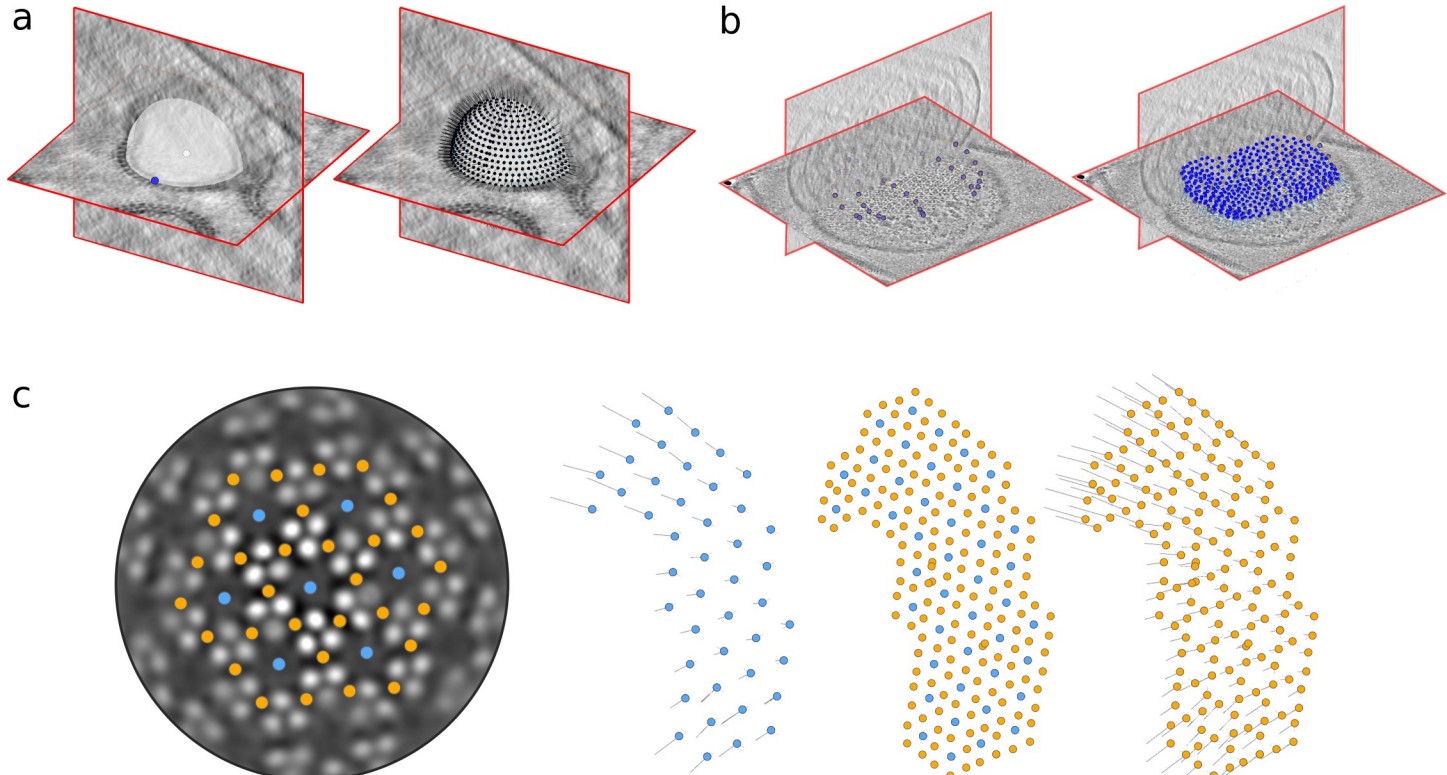

**Fig 4. Geometrical approaches to STA.** Particle positions and orientations are generated from minimal manual annotations. In (a), a sphere is defined by a center and edge point, and positions with an orientation normal to the surface are seeded on the sphere. In (b), an arbitrary surface is defined by annotation of points on that surface in a number of tomographic slices. A supporting mesh is derived from these points on which a set of particle positions and orientations are seeded, again with an orientation normal to the surface. (c) Different positions in a chemosensory array (EMD-10160; see also below) are annotated in blue and orange. A set of particle poses from an STA experiment (blue) is used to derive the positions and orientations of the orange particles in a "subboxing" procedure. STA, subtomogram averaging.

## Providing access to alternative refinement and classification procedures

*RELION* implements a Bayesian approach to the refinement of one or many 3D volumes from cryo-EM data called maximum a posteriori refinement [22]. Originally designed for single-particle cryo-EM data and more recently extended to work with cryo-ET data [32], *RELION* provides solutions for the "refinement" block of the cryo-ET pipeline. The package features an easy-to-use graphical user interface (GUI) and is already in use in a large number of cryo-EM labs. Rather than a data processing pipeline, *Dynamo* is a collection of powerful tools for working with STA data. With an emphasis on customisation, *Dynamo* often exposes the user to large numbers of parameters in fully featured GUIs.

By interfacing *Dynamo* with the *Warp-RELION-M* pipeline in the "refinement" block, we provide *Warp-RELION-M* users access to highly customisable STA and multireference classification procedures, principal component analysis (PCA)-based classification, and myriad data visualisation tools and analysis tools. *Dynamo* users gain a route to making use of the CTF estimation and particle reconstruction tools in *Warp*, STA and classification in *RELION*, and the multi-particle refinement procedure implemented in *M*.

As an example of a possible benefit of increased flexibility in STA workflows, we compare the options for masking during iterative refinement in *Dynamo* and *RELION*. In a 3D classification experiment using *relion_refine*, one mask is used for alignment, classification, and Fourier shell correlation (FSC) for the determination of regularisation parameters between iterations. In a *Dynamo* multireference classification project, a user may choose to provide separate masks for these 3 procedures. As an example of the possible benefits, this allows for the optimisation of particle alignment parameters during a focused classification experiment. In addition, creative combination of masks can allow for exclusion of a membrane from alignment for a target that would otherwise require using masks with extremely soft edges to avoid FSC artefacts. Such a soft mask would necessarily include the membrane or exclude membrane-proximal protein density.

## Application to the HIV-1 CA-SP1 hexamer (EMPIAR-10164)

We illustrate the power of working within this framework on an example dataset by reprocessing a 5 tilt-series subset of *EMPIAR-10164* containing immature HIV-1 dMACANC virus-like particles (VLPs). This dataset was contributed to the community by the Briggs group, and this subset has previously been used to benchmark *NovaCTF* [17] and *Warp* [10], resulting in 3.9 Å and 3.8 Å reconstructions, respectively.

Initial motion correction and initial CTF estimation were performed in *Warp* with respective spatiotemporal model resolutions of (1, 1, 8) and (2, 2, 1). Tilt series were automatically aligned using *autoalign_dynamo* before CTF estimation and tomogram reconstruction at 10 Å/px in *Warp*. VLPs were annotated in *Dynamo*, and initial estimates of positions and orientations were generated with an interparticle distance of 45 Å, oversampled relative to the approximately 75 Å lattice spacing seen in the tomograms. A total of 500 particles were extracted and averaged in *Dynamo* to produce a coarse template. The same data were subject to STA against this template in *Dynamo* without imposing symmetry during refinement. The resulting average contained a hexagonal lattice. To obtain an initial model, the 6-fold axis of the lattice was aligned to the z-axis, and the resulting volume was symmetrised 6-fold in *Dynamo*. A total of 28,516 particles were extracted from annotated VLPs and aligned against the initial model for one iteration with a limited angular search range in *Dynamo*. The resulting particle positions and orientations, visualised in *Dynamo*, formed regular lattices on the surfaces of each VLP with some less regular areas. Using simple MATLAB scripts, only the 19,810 particles with 3 or more neighbours at the expected interparticle distance were retained for subsequent analysis.

Using *dynamo2m*, metadata were converted to enable working in *Warp-RELION-M*. Particle reconstructions were carried out in Warp and 3D refinements in *RELION 3.1*, starting with local angular searches of ±15°. Resolution estimates were measured by FSC between independent reconstructions from random half-sets of the data using *relion_postprocess*. Particles were (i) reconstructed at 5 Å/px and refined (unmasked) to 10 Å resolution; (ii) reconstructed at 2.5 Å/px and refined (unmasked) to 5 Å resolution; and (iii) reconstructed at 1.7 Å/px in *Warp* and refined (masked to include only the central hexamer) to 3.8 Å resolution. This intermediate result is indicative of the accuracy of fiducial-based alignment workflows in *Dynamo* and alignments in *RELION*. Seven rounds of multi-particle refinement were performed in *M* at 1.6 Å/px in a masked region containing only the central hexamer. The first 4 rounds of multi-particle refinement, optimising only for deformation parameters, yielded a 3.6-Å reconstruction. Three further rounds including optimisation of electron-optical aberration–related parameters and tilt frame alignments refined to a resolution of 3.4 Å (Fig 5).

It should be noted that working within this framework readily yielded a 3.4-Å reconstruction from 19,810 particles in 5 tomograms, without use of an external reference and leveraging the geometry of the system evidenced in the tomograms during alignments. The combination of *Dynamo* tools and the *Warp-RELION-M* pipeline, enabled by *autoalign_dynamo* and *dynamo2m*, rendered this ab initio, geometrical approach a smooth, efficient process. An ab initio approach is not strictly necessary for this dataset, as demonstrated by *emClarity* [7] and *Warp-RELION-M* [9]. We, however, note that obtaining an initial model is often a challenging step in the first stages of a project. Thus, access to ab initio and geometrical approaches is provided as an alternative, should the integrated template matching procedure prove inadequate. The final resolution of the reconstruction serves only to demonstrate the ability of *M* to further optimise image alignment and electron-optical parameters.

## Application to the *Escherichia coli* chemosensory array in situ

We are currently using an optimised minicell system [33] to investigate structural bases of the chemotactic response in motile bacteria. To illustrate the advantages of working within this framework, we choose to present intermediate results from a set of 109 minicell tomograms (Fig 6).

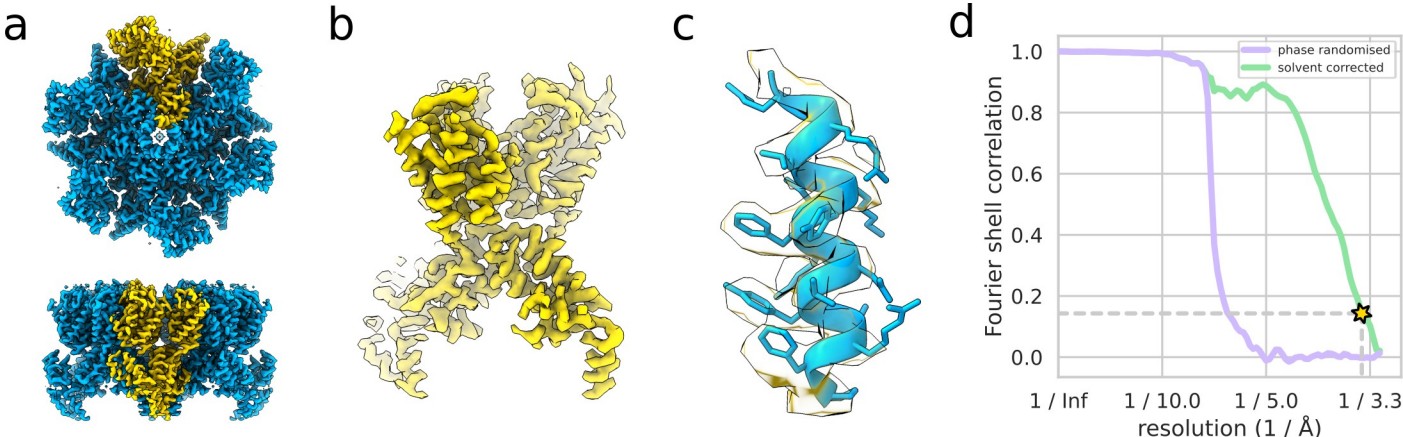

**Fig 5. A (nominally) 3.4-Å reconstruction of the HIV-1 CA-SP1 hexamer from 19,810 particles in 5 tilt series of *EMPIAR-10164*.** The whole hexamer with one monomer highlighted in yellow (a), one monomer (b), and the region of the map around helix 292–307 PDB 5L93 (fitted) (c) are shown. The gold-standard FSC curve (d) used to estimate the nominal resolution value is shown as well as the phase-randomised masked FSC used to validate the mask used for FSC calculation. The data used to produce this figure can be found at http://doi.org/10.5281/zenodo.4783129. FSC, Fourier shell correlation.

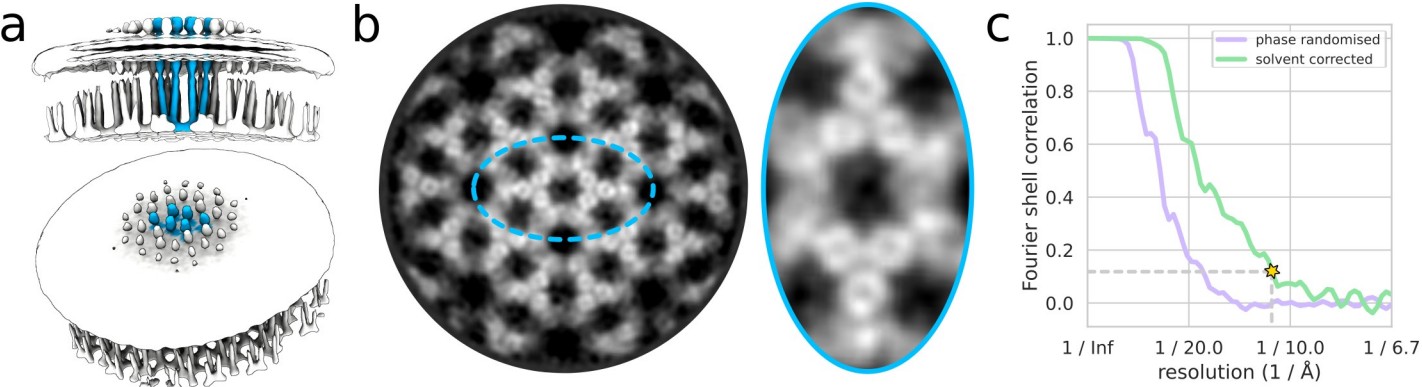

**Fig 6. A (nominally) 11-Å reconstruction of the in situ chemosensory array from 17,484 particles in 48 tomograms of *E. coli* minicells.** The full reconstruction with a pixel size of 7.5 Å/px is shown in (a) with the membrane density superposed at a lower isosurface threshold. The quality of the density is visible in a slice through the map reconstructed with a pixel size of 2.24 Å/px and filtered to 11 Å (b, left). Holes are clearly visible in the densities of receptor dimers at this resolution (b, right). The gold-standard FSC curve (c) used to estimate the nominal resolution value is shown alongside the phase-randomised masked FSC curve used to validate the mask used for FSC calculation. The data used to produce this figure can be found at http://doi.org/10.5281/zenodo.4783151. FSC, Fourier shell correlation.

The chemotaxis-mediating molecular machinery forms large supramolecular arrays, a highly cooperative network of core signalling units (CSUs), in the bacterial inner membrane at the cell poles. The elongated nature of the CSU, the crowded cellular milieu, and increased sample thickness complicate template matching approaches with this system, increasing the false-positive rate and impeding appropriate peak extraction. Instead, we opt for a seed-over-sampling approach in which initial positions and orientations are distributed on *Dynamo* surface models, oversampling with respect to the expected interparticle distance of approximately 120 Å. Particle positions were generated from the surface model with an interparticle distance of 30 Å using *Dynamo*. Particles were extracted and subject to one round of STA in *Dynamo* with local angular searches of ±30°. Constraining the angles of particles relative to initial estimates from a supporting geometry ensures that particles within each array have orientational parameters coherent with our understanding of the system. Particles were extracted at 7.5 Å/px in *Warp* and imported into *Dynamo* using *warp2dynamo*.

A consensus refinement of these particles, centred on the 3-fold symmetry axis of a p6-symmetric assembly, was performed in *Dynamo*, allowing local deviations (±30°) in out-of-plane orientation, complete search of in-plane orientation, and enforcing C3 symmetry during refinement. In the resulting average, the expected 3-fold symmetric arrangement of P4 domains in the structure was not readily observed. The resulting reconstruction had a nominal resolution of 18 Å, measured by gold-standard FSC, with the FSC curve indicating the presence of significant heterogeneity. The *Dynamo* tool *dpktbl.subbox.tableOnTable* was used to derive the positions and orientations of particles centred on 6 intertrimer-of-dimer axes of the array. Duplicate particles were removed using *dpktbl.exclusionPerVolume* prior to a consensus refinement of these particles enforcing C2 symmetry. The resulting reconstruction had a nominal resolution of 19 Å. Attempts to disentangle structural heterogeneity using multireference approaches to 3D classification in a variety of masks using both *Dynamo* and *RELION* failed to yield sensible results at this stage.

Multi-particle refinement in *M* yielded improvements when performed in a masked region encompassing a large array region of 400 Å diameter, resulting in a reconstruction with a nominal resolution of 11 Å. No improvement was seen when refinement was performed on smaller regions encompassing 1 to 4 CSUs, presumably due to insufficient signal for accurate alignments. Work is underway to improve upon this intermediate result by classification and

further refinement. However, these initial results are already quite promising, showing features such as a hole in the center of the 4-helical bundle of a receptor dimer, not yet seen in situ.

## A comprehensive guide to obtaining a 3.4-Å reconstruction of the HIV-1 CA-SP1 hexamer

Information about best practices and approaches to problems presented by cryo-ET data analysis is constantly evolving and currently divided between the literature, mailing lists, and documentation specific to various pieces of software. We provide, in the form of a rich, living document, a step-by-step walk-through for the procedure used to obtain the 3.4-Å reconstruction of the HIV-1 CA-SP1 hexamer (https://teamtomo.org/walkthroughs/EMPIAR-10164/introduction.html). This guide takes a user through downloading a 5 tilt series subset of *EMPIAR-10164*, preprocessing multiframe micrographs in *Warp*, aligning tilt series automatically using *autoalign_dynamo*, generating tomograms in *Warp*, setting up a *Dynamo* catalogue for geometrical particle picking using *warp2catalogue*, picking particles based on supporting geometries in *Dynamo*, initial model generation in *Dynamo*, isolating an optimal subset of particles using custom scripts, reconstruction of particles in *Warp* via *dynamo2m*, rigid body STA in *RELION*, and multi-particle refinement in *M*. Working through this guide is designed to expose a user to the theory and practice of obtaining a reconstruction from cryo-ET data. They will learn to use a variety of computational tools to address problems they will likely encounter when analysing their own data. The guide is provided as part of https://teamtomo.org/, an open collaborative platform we established for sharing knowledge in the growing cryo-ET community. Those wishing to share their expertise for the benefit of the wider community are invited to contribute.

## Conclusions

The tools and resources presented in this manuscript increase automation within the *Warp* preprocessing pipeline, simplify metadata handling, and interface two complementary cryo-ET data processing packages. The interface provided by *dynamo2m* enables working within a powerful, flexible framework for geometrical and ab initio approaches to STA, which can benefit from multi-particle refinement in *M*. Our tools increase the functionality available to users of both *Dynamo* and *Warp-RELION-M*, lowering the barrier to employing more customisable workflows for cryo-ET data processing, which are often key to success. Finally, we have attempted to make structural cryo-ET more accessible by providing a comprehensive, step-by-step walk-through to obtaining state-of-the-art STA results. It is our hope that this online guide can serve as a starting point for newcomers to the growing cryo-ET community, enabling them to more quickly understand and use cryo-ET to solve their specific biological questions. Furthermore, we anticipate that the existence of a collaborative, community-driven resource for sharing cryo-ET knowledge and tools will encourage open science and contribute to the development of cryo-ET as a valuable tool for structural cell biology.

## Supporting information

**S1 Fig. The number of STA reconstructions with different reported resolution values in the EMDB.** The data used to produce this figure can be found at https://www.ebi.ac.uk/pdbe/emdb/statistics_main.html. EMDB, Electron Microscopy Data Bank; STA, subtomogram averaging.
(TIF)

**S2 Fig. The source code for generating the scene depicted in Fig 2, combining starfile and eulerangles with existing tools in the scientific Python ecosystem.**
(TIF)

## Acknowledgments

We acknowledge Diamond Light Source for access and support of the cryo-EM facilities at the UK's national Electron Bio-imaging Centre (eBIC), funded by the Wellcome Trust, MRC, and BBRSC. Cryo-ET data acquisition has been supported by iNEXT, grant number 653706 (PID:2626 to IG), funded by the EU Horizon 2020 programme.

## Author Contributions

**Conceptualization:** Alister Burt, Irina Gutsche.

**Data curation:** Alister Burt.

**Funding acquisition:** Irina Gutsche.

**Investigation:** Alister Burt, Lorenzo Gaifas, Tom Dendooven, Irina Gutsche.

**Methodology:** Alister Burt, Irina Gutsche.

**Project administration:** Irina Gutsche.

**Resources:** Irina Gutsche.

**Software:** Alister Burt.

**Supervision:** Irina Gutsche.

**Validation:** Alister Burt, Lorenzo Gaifas, Tom Dendooven, Irina Gutsche.

**Visualization:** Alister Burt.

**Writing – original draft:** Alister Burt, Irina Gutsche.

**Writing – review & editing:** Alister Burt, Lorenzo Gaifas, Irina Gutsche.

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
