## [Editor Report · Decision Letter 0]

9 Mar 2021

Dear Dr Gutsche, 

Thank you for submitting your manuscript entitled "Tools enabling flexible approaches to high-resolution subtomogram averaging" for consideration as a Method and Resources paper by PLOS Biology. Please accept my sincere apologises for the great delay in the processing of your manuscript as we consulted with an academic editor about your submission. Your manuscript has been considered as a dual submission with PBIOLOGY-D-21-00587. This decision letter is in regard to to PBIOLOGY-D-21-00462 and you will receive a separate independent decision letter for PBIOLOGY-D-21-00587.

Your manuscript has now been evaluated by the PLOS Biology editorial staff, as well as by an academic editor with relevant expertise and I am writing to let you know that we would like to send your submission out for external peer review.

Please note, however, that the outcome of our discussion of your manuscript is that we have some reservations as to the overall strength of advance provided by your manuscript over existing methodological approaches. We would need to be persuaded by the enthusiasm of the reviewers that the paper has the potential after revision to offer the significant strength of advance that we require for publication in order to pursue it further for PLOS Biology.

Please re-submit your manuscript within two working days, i.e. by Mar 11 2021 11:59PM.

Kind regards,

Richard

Richard Hodge, PhD

Associate Editor, PLOS Biology

rhodge@plos.org

---

## [Decision Letter · Decision Letter 1]

12 Apr 2021

Dear Dr Gutsche,

Thank you very much for submitting your manuscript "Tools enabling flexible approaches to high-resolution subtomogram averaging" for consideration as a Methods and Resources article at PLOS Biology. Your manuscript has been evaluated by the PLOS Biology editors, an Academic Editor with relevant expertise, and by several independent reviewers.

The reviews are attached below. You will see that the reviewers find your integrated software workflow for subtomogram averaging to be accessible and very useful for the field, but they also raise concerns with the presentation and reporting of the workflow in the manuscript. We ask that you please take into consideration the comments of Reviewer #3 to restructure and reorganise the revised manuscript. 

In light of the reviews, we will not be able to accept the current version of the manuscript, but we would welcome re-submission of a revised version that takes into account the reviewers' comments. We cannot make any decision about publication until we have seen the revised manuscript and your response to the reviewers' comments. Your revised manuscript is also likely to be sent for further evaluation by the reviewers.

We expect to receive your revised manuscript within 1 month.

**IMPORTANT - SUBMITTING YOUR REVISION**

*Resubmission Checklist*

*Published Peer Review*

*PLOS Data Policy*

*Blot and Gel Data Policy*

Sincerely,

Richard

Richard Hodge, PhD

Associate Editor, PLOS Biology

rhodge@plos.org

REVIEWS:

Reviewer's Responses to Questions

PLOS authors have the option to publish the peer review history of their article (what does this mean?). If published, this will include your full peer review and any attached files.

Reviewer #1: No

Reviewer #2: No

Reviewer #3: No

Reviewer #1: In this manuscript, the authors describe an integrated workflow for subtomogram averaging by combining Dynamo with the recently published Warp-RELION-M pipeline. The workflow allows simplified metadata handling, increased automation, and flexible procedure. Moreover, the authors establish an open and collaborative online platform for sharing cryo-ET data analysis knowledge. The authors identify the problem that, while subtomogram averaging processing has developed alongside single particle analysis and is becoming increasingly more powerful, the software technology is fragmented, and users often have to become creative in choosing how they want to process and analyze data. This makes it difficult particularly for newcomers to cryo-ET.

Overall, this work will be very useful to the field of cryo-ET. Seeing all the different customization tools in Dynamo combined with the powerful refinement of Warp-RELION-M is really exciting. The online platform is a great addition that will hopefully help increase communication among users in the field and allow continued improvement of the methods. 

Questions and comments:

1. Lines 106-148: How does the accuracy of Dynamo fiducial tracking and tilt series alignment compare with that was done with IMOD alone? 

2. Lines 185-205: The authors highlight an important point: metadata handling can be a pain. Aside from a nice visualization of particle poses in Napari, I wonder if there is anything else they can show in this results section to guide the user (especially a newcomer) through the use of the different packages that they discuss. For example, a flowchart to go with the source code provided.

3. Lines 240-242: Some clarity or description of how these geometric orientations were generated in Dynamo would be helpful. For example, for figure 3a I assume that the virus membrane was segmented, and particles were oriented orthogonal to the membrane surface. However, in 3b it is unclear to me how those orientations were produced. 

4. Lines 293-324: It will be very beneficial to show a flowchart of the detailed processes used, along with stepwise results shown, so the readers can have a better understanding and expectation of what to produce while trying the workflow on the VLPs or similar datasets. 

5. Line 299: What's the method for CTF correction in this proposed workflow, and how does it compare to NovaCTF used in the referenced VLP STA paper?

6. Lines 305-306: The alignment parameters (angular search, steps, etc.) should be at least briefly mentioned here. Same for the lines 314-316.

7. Lines 308-316: If an average was produced in Dynamo with 19,000 - 26,000 particles, why is it also beneficial to do more refinement in RELION? The authors should emphasize why using RELION is better than using only Dynamo. This could be done potentially by comparing resolutions of final averages without using RELION and only using Dynamo, Warp, and M. A clear explanation of the advantages of RELION over Dynamo for these steps will be helpful.

8. Lines 359-363: Why were particles extracted in Dynamo for an initial template, and then further extracted from Warp and brought into Dynamo?

9. Lines 361-375: Again, similar to the comment #4 above, showing the workflow and intermediate results in the form of a flowchart would be very beneficial for new users to learn/try this method on similar data. 

10. Line 373-375: What exactly were attempted to disentangle structural heterogeneity and how did they fail?

11. At the end, a detailed section on hardware requirements and how to connect the systems/computers together (i.e., the need for Linux vs. Windows operation systems, CPU/GPU requirements, how to properly connect these programs to one another locally or over the network, etc.) would be critical for new users to set the system up. 

12. A note on computing time took for different steps in the workflow of the reported two example projects, with information of what computation power were used, would be beneficial for new users to plan for setting up the system.

Reviewer #2: Summary: Burt et al describe new software packages facilitating data

transfer between Dynamo, M, Relion, and Warp and increased automation.

Use of these tools is illustrated via application to EMPIAR-10164 and

an E. Coli chemosensory array. Increased integration between these

packages and the associated flexibility will certainly be appreciated

by users. Generally the paper is clear and well-written. If I were to

restrict myself to a single, high-level comment, it would be that more

explicit, tutorial examples would be helpful, but perhaps the authors

have choosen to provide that elsewhere. See below for more specific

comments.

1) Lines 84-86: I suggest being more explicit about the facts that the

final stages of "modern" STA refinement has more in common with SPA than

with classical SPA, and that tilt series alignments and subvolumes are

no longer considered fixed , but rather as initial estimates subject to

further optimization.

2) Lines 109-119: IMOD, Dynamo, and Warp all contribute to your tilt

series alignment, but the roles and contributions of each package have

not been made clear. Automated alignment has supposedly been available

in IMOD since 2017 (Mastronarde and Held JSB 197:102-113). At the very

least, this paper should be cited, and the various approaches compared

and contrasted. Describe why you think what you've done is an

improvement over previous work.

3) Line 149: Euler angle conversion routines are useful, but I question

the benefit of yet another such package. Several are already available.

E.g. in MATLAB the Aerospace Toolbox provides euler2dcm and dcm2euler.

In Python, packages scipy.spatial.transform and Pspincalc provide

similar functionality. Moreover, in my opinion the fundamental issue

is not Euler angle conversions per se, which are routine, but rather the

lack of full documentation on conventions used by various EM packages.

CCP-EM has attempted to summarize these, but their list is neither

sufficiently complete nor always accurate, since it relies on the

packages internal documentation. It would be extremely helfpul if

someone would verify and summarizing complete rotation convention

information for popular EM packages. Even assuming a right handed

coordinate system, such a description must include:

A) Is the right-handed or left-handed screw rule used?

B) Are transformations active or passive?

C) Are Euler angles intrinsic or extrinsic?

D) Is the default transformatio reference-to-particle or vice-versa?

and, finally, of course,

E) The Euler or Tait-Bryan / Cardan sequence used.

4) Lines 181-183: Why provide relion_star_downgrade, rather than

updating Warp to (optionally) support the Relion 3.1 star format?.

Few users should still be using Relion 3.0, and you don't yourselves!

5) Lines 198-199: Agreed, but many other packages already provide such

exploratory data analysis capabilities. Dynamo, for example, already has

access to a full range of EDA tools in MATLAB.

6) Lines 275-276: It's not strictly true that Relion uses the same mask

for alignment, classification, and FSC's. During 3D refinement, Relion

will use the same mask for alignment and FSCs, and also true that during

2D or 3D classification, if alignment is being done the same mask will be

used for alignment and classification. However there is no requirement

that the mask used during classification is the same as that during

refinement or that either is the same as the mask used during final FSC

computation or post-processing. Relion already provides most of the

flexibility as you're describing.

7) Figure 4: This figure and it's description could be improved. No

offense, but your logos are hardly universally recognized. Many readers

will not realize that a gear = Dynamo, a W = Warp, and M = M. Label them,

add a legend, or make the description more complete.

8) Line 386: Rather than just "teamtomo.org", I suggest listing as a

fully qualified url (and, of course, as a functional link in electronic

versions).

Reviewer #3: Summary of key results:

This paper provides a comprehensive workflow for solving structures using subtomogram averaging from cryo-electron tomography data. The authors build upon algorithmic approaches already established in the field, and develop a series of scripts to automate and facilitate the process of interacting between various programs. The end result is a computational framework to make it easier to process tilt series, reconstruct tomograms, pick particles, and incorporate a priori orientation information calculated from multiple sources to facilitate ab initio subtomogram averaging. This workflow is benchmarked on two datasets of membrane-associated protein complexes, the HIV-1 CA-SP1 hexamer and chemosensory arrays, yielding results both comparable to previous reports and with some additional novel structural insights. The authors also establish an open-access and collaborative platform to disseminate their pipeline and to encourage other tomographers to contribute their respective protocols outlining various steps of the cellular tomography workflow. 

Strengths, originality, and significance:

It is often challenging for newcomers, particularly those lacking extensive scripting and programming skills, to navigate through the tomography reconstruction and subtomogram averaging pipeline to produce a high resolution and/or biologically-interpretable structures. Although there have been some efforts in recent years by groups to alleviate this steep learning curve, these workflows are often restricted to a single software package without much flexibility to navigate and utilize beneficial aspects of multiple algorithmic approaches. If we want to expand the user-base of cryo-electron tomography to both structural and cellular biologists alike, we need to develop more streamlined and user-friendly workflows. This manuscript explicitly addresses this need and opens up an exciting option for the collective tomography community to take part in this effort through an open-access online platform. 

Suggested improvements:

Overall, the content of the manuscript is excellent and the streamlined approach is very much needed in the field. However, the overall format, organization, and presentation of the manuscript currently comes across a bit clunky and non-linear. It tends to jump around between different steps, providing details in some sections, and then again more details in other sections (e.g. explanation of Euler angles, mention of scripts in both the methods and results/discussions section, etc.). The manuscript would be greatly improved through restructuring and reorganization, with the goal in mind of it being used primarily (and in practice) as a protocol for subtomogram averaging. One suggestion would be to restructure the manuscript with the target audience in mind, which is most likely a user that has either (1) just started their structure-solving journey (with little to no experience) or (2) tried using a single program and was left frustrated that initial attempts failed to produce an interpretable structure. In both cases, the audience likely has some knowledge of the process, but are searching for a "how-to" manual that will guide them through the (often confusing) process. 

Here are a few actionable suggestions:

1. Reformat manuscript to have more of a protocol format. Perhaps sections for the manuscript could include the following: Introduction, Overview of approach, application of methodology, Conclusion.

2. The introduction generally reads a bit clunky and does not build enough rationale for the described workflow. As someone who has experienced firsthand the frustration and annoyance of navigating through the various software packages, the rationale is very obvious to me, and is why I believe the content of this study is particularly important and relevant for the field. However, I worry that without this "insider" knowledge, the importance of this work might be missed. Perhaps better to build rational by listing all of the various steps needed for tomogram reconstruction and subtomogram averaging, and give examples to illustrate why it is difficult to integrate and interact with all of them (e.g. different file inputs and outputs, etc.). For example, lines 258-279 are great rationale and would be more appreciated in the introduction. 

3. "Overview of approach" could contain sub-sections that go over various steps, in order. Content-wise, this would include everything from "materials and methods" through "framework for flexible ab initio...". It would be better to re-format as a protocol for the workflow, with links to the various scripts embedded in the text. Lines 299-324 read more like a methods section, and it would be useful to include in the "overview of approach" or perhaps even in a supplemental methods section.

4. It would be helpful to adapt figure 4 to make it explicit which scripts are novel to this work, and which are already established. Some of the scripts are shown, but not all of them, and it gets a bit lost which ones were developed in this study.

---

## [Editor Report · Decision Letter 2]

9 Jun 2021

Dear Irina,

On behalf of my colleagues and the Academic Editor, Grant Jensen, I am pleased to say that we can in principle offer to publish your Methods and Resources article "A framework for ab initio and geometrical approaches to multi-particle refinement in cryo-electron tomography" in PLOS Biology, provided you address any remaining formatting and reporting issues. I have taken over the handling of your manuscript during the absence of my colleague, Richard Hodge, from the office to prevent unnecessary loss of time.

We would like to once again raise the issue of the title, as titles are the most important determinant that will influence the readership of your article. It needs to be informative of course, but also accessible to those that would benefit from the framework that you put forward and who may not be very familiar with the field's jargon. In this regard, "ab initio and geometrical approaches to" are quite technical and will not be meaningful to a subtantial fraction of potential readers. As the specifics of the framework are very clear in the abstract, we would strongly recommend a more approachable title. Perhaps the following would be a good compromise?

"Computational framework to streamline multi-particle refinement in cryo-electron tomography" 

(For context, the final title of the Castano-Die et al work will be "Step-by-step guide to efficient subtomogram averaging of virus-like particles with Dynamo").

In addition to the change in title, any outstanding formatting or reporting issues that need to be addressed before publication will be detailed in an email that will follow this letter and that you will usually receive within 2-3 business days, during which time no action is required from you. Please note that we will not be able to formally accept your manuscript and schedule it for publication until you have made all the required changes.

PRESS

Thank you again for choosing PLOs Biology for publication and supporting Open Access publishing. We look forward to publishing your paper. 

Sincerely, 

Nonia

Nonia Pariente, PhD

Editor in Chief 

PLOS Biology

npariente@plos.org